# Molecular-resolution imaging of ice crystallized from liquid water by cryogenic liquid-cell TEM

Jingshan S. Du [1], Suvo Banik [2,3], Henry Chan [2], Birk Fritsch [4]
Ying Xia [5], Ajay S. Karakoti[1], Andreas Hutzler [4],
Subramanian K. R. S. Sankaranarayanan [2,3] & James J. De Yoreo [1,5] ✉

Despite the ubiquity of ice, a molecular-resolution image of nanoscopic defects or microstructures in ice crystallized from liquid water has never been obtained. This is mainly due to the difficulties in preparing and preserving crystalline ice samples that can survive under high-resolution imaging conditions. Here, we report the stabilization and Å-resolution electron imaging of ice $I_h$ crystallized from liquid water by developing cryogenic liquid-cell transmission electron microscopy (CRYOLIC-TEM). We combine lattice mapping with molecular dynamics simulations to reveal that ice formation is highly tolerant to nanoscale defects such as misoriented subdomains and trapped gas bubbles, which are stabilized by molecular-scale structural motifs. Importantly, bubble surfaces adopt low-energy nanofacets and create negligible strain fields in the surrounding crystal. These bubbles can dynamically nucleate, grow, migrate, dissolve, and coalesce under electron irradiation and be monitored in situ near a steady state. This work improves our understanding of water crystallization behaviors at a molecular spatial resolution.

Ice crystallization is one of the most important processes in the ecosphere and is central to atmospheric processes[1–3], transportation safety[4–7], biomedical cryopreservation[8,9], and the food industry[10,11]. Among all ice species, hexagonal ice (type $I_h$) crystallized from liquid water is most prevalent in the ambient. Due to the weak hydrogen bonds between water molecules, these structures can easily deform on the molecular scale during crystallization[12,13]. The precipitation of dissolved gas in water can further generate cavities in ice crystals; their formation and migration are exclusive to liquid-crystallized ice[14,15] with profound implications in glaciology and paleoclimatology[16,17]. Elucidating defects and microstructures of ice, particularly their molecular origins, is critical to understanding the thermodynamics, phase transformation, and mechanical properties of ice[18] and many other hydrogen-bonded crystals[19–21].

Despite substantial interest in ice on the molecular scale, most studies rely on in-silico simulations[22–25] and ensemble-scale spectroscopy and diffraction[26–28]. Meanwhile, real-space imaging in ice at this scale remains incredibly challenging due to the conflict between the low stability of ice and the harsh and invasive conditions often required by high-resolution imaging techniques. Recent breakthroughs in low-dose cryogenic transmission electron microscopy (cryo-TEM)[29–32] and ultra-high vacuum scanning probe microscopy[33,34] have enabled atomic-resolution imaging of ice condensed from the gas phase or converted from vitrified films. However, these experiments have typically been limited to crystals with random shapes and far-from-equilibrium structures because they rely on phase transformation and deposition at ultra-low temperatures in a high vacuum. Although cryo-TEM has recently been used to

[1]Physical Sciences Division, Pacific Northwest National Laboratory, Richland, WA, USA. [2]Center for Nanoscale Materials, Argonne National Laboratory, Lemont, IL, USA. [3]Department of Mechanical and Industrial Engineering, University of Illinois, Chicago, IL, USA. [4]Helmholtz Institute Erlangen-Nürnberg for Renewable Energy (IET-2), Forschungszentrum Jülich GmbH, Erlangen, Germany. [5]Department of Materials Science and Engineering, University of Washington, Seattle, WA, USA. ✉e-mail: james.deyoreo@pnnl.gov

observe crystalline ice films prepared by liquid nitrogen, the structures that have been observed are limited to single crystals and low-energy stacking faults[35]. Molecular-resolution imaging of the microstructures and defects formed by the crystallization of liquid water remains elusive to date.

Herein, we report an approach to freeze liquid water into high-quality ice $I_h$ samples, termed cryogenic liquid-cell TEM (CRYOLIC-TEM), which allows for high-resolution TEM (HRTEM) imaging. Inspired by advances in liquid-phase electron microscopy[36], this method freezes liquid water between amorphous carbon (a-C) membranes into large-area (up to microns in size) single-crystalline ice $I_h$ films stable under the electron beam. Aberration-corrected HRTEM imaging at a line resolution better than 2 Å is routinely achieved with a record of 1.3 Å in continuous single-crystalline regions. This new capability enables us to directly correlate lattice mapping with molecular dynamics (MD) simulations based on machine-learned models to elucidate defect nanostructures formed by liquid water crystallization. The formation, migration, coalescence, and dissolution trajectories of nanobubbles in single-crystalline ice are further observed in situ under the electron beam. This work provides a versatile way for accessing close-to-equilibrium ice $I_h$ in TEM and unlocks previously inaccessible avenues to probe the nano- and molecular-scale interfacial configurations of ice and their structural dynamics.

## Results

### Stabilizing ice $I_h$ single crystals from liquid water

To freeze liquid water into ice $I_h$ membranes suitable for TEM imaging, we first encapsulated deionized water between two TEM grids coated with a-C membranes and then loaded the sample onto a cryo-TEM sample holder (Fig. 1a). The sample was subsequently cooled by liquid nitrogen along with the holder. This process is substantially slower than vitrification in standard cryo-TEM, allowing for the crystallization of water. Flat, robust, and smooth a-C membranes are necessary for obtaining large-area, high-quality ice single crystals (Supplementary Fig. 1). The surface properties of the a-C membrane are similar to that of graphite (Supplementary Figs. 2 and 3). Exposing the cold sample to the atmosphere may result in the condensation of ice spherulites on the a-C film. This low-temperature (typically < −180 °C) gas-phase ice deposition results in far-from-equilibrium mixtures of ice $I_h$ and $I_c$ (Supplementary Fig. 4), consistent with literature reports[26,30,31,37]. In contrast, ice formed by water crystallization between the a-C membranes often shows a flat morphology (Fig. 1b) and can be easily distinguished from condensed ice. In these crystals, single-crystalline regions aligned (or close) to the [0001] zone axis can often be found with areas up to several microns (Fig. 1c and Supplementary Figs. 5 and 6). The thickness of encapsulated ice can vary from ~10 nm to micron-scale, depending on the geometry of the pocket (see Supplementary Fig. 9). Still, high-quality HRTEM images can be acquired

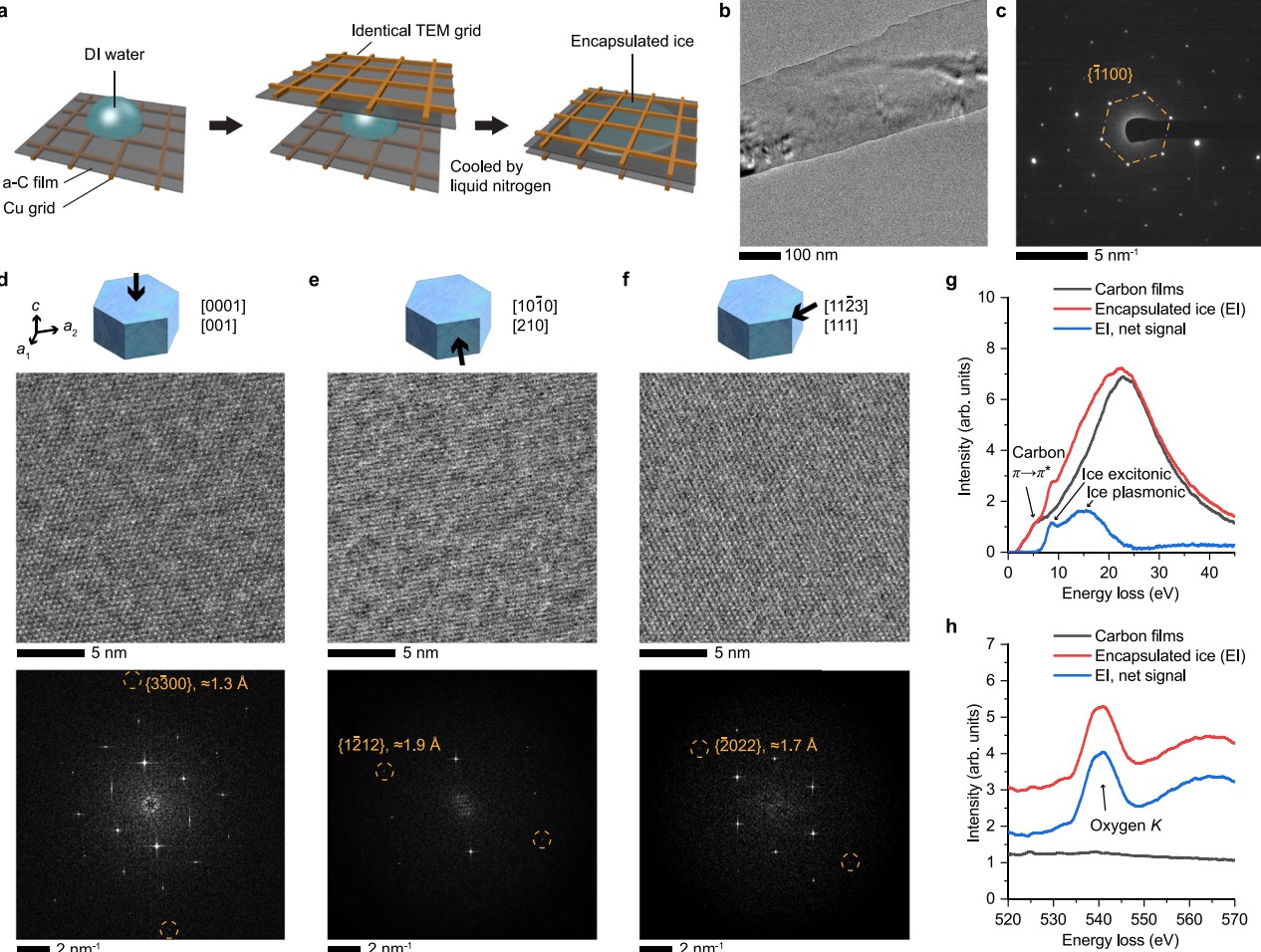

**Fig. 1 | Crystallization of liquid water for HRTEM. a** Schematic of encapsulating ice crystal sections from deionized (DI) water between amorphous carbon (a-C) films. **b** TEM image of a thin strip of ice crystal encapsulated between a-C films. **c** SAED pattern showing overall single crystallinity along the [0001] zone axis. **d**–**f** Schematic of the viewing direction (top row), average background subtraction (ABS)-filtered HRTEM (middle row), and Fourier transform (bottom row) of ice $I_h$ along three zone axes. Both Miller-Bravais and Miller indices were given for convenience. EELS profiles in the low-loss (**g**) and oxygen $K$ core-loss (**h**) regions. The low-loss profiles were deconvolved using the Fourier-log algorithm to remove the zero-loss peak and plural scattering. Source data are provided as a Source Data file.

from relatively thin regions, achieving a record line resolution of 1.3 Å (Fig. 1d). Preparing samples on a double-tilt cryo holder further facilitates diffraction and HRTEM imaging of single crystals along other zone axes (Fig. 1e, f, and Supplementary Figs. 7 and 8). Importantly, these single-crystalline areas can withstand HRTEM imaging for at least minutes (electron flux up to ~100 e Å$^{-2}$ s$^{-1}$) without developing discernable nanoscale defects (Supplementary Fig. 10 and Supplementary Movies 1 and 2).

Cryogenic electron energy-loss spectroscopy (EELS) was performed to evaluate the high chemical purity of the encapsulated ice (Fig. 1g, h). After subtracting the inelastic scattering from the a-C membranes, the low-loss spectrum of encapsulated ice shows a relatively sharp peak centered at ≈8.5 eV and a broad peak at ≈15 eV (Fig. 1g). The former is characteristic of the excitonic $1b_1 \rightarrow 4a_1$ orbital transition of $H_2O$ in ice[38,39] and can be unequivocally distinguished from the $\pi \rightarrow \pi^*$ transition from graphitic carbon[39] or aromatic organics[38] at lower energy. Accordingly, this method results in highly pure ice samples by avoiding the organic contamination frequently encountered by other encapsulation techniques, such as frozen graphene liquid cells[40], caused by the solute concentration effect[41,42]. The broad peak is located at a lower energy than the bulk plasmon typically reported in the literature[29,38,39], possibly due to the ice-carbon interface. Core-loss EELS profiles for the oxygen $K$ edge also confirmed the highly pure chemical environment of the encapsulated ice (Fig. 1h).

## Nanoscale subdomains at a defective crystal edge

The strip-shaped crystal section in Fig. 1b appears to be a single crystal according to diffraction criteria (Fig. 1c). Surprisingly, lattice-resolved HRTEM images reveal highly defective textures near the edge areas despite a perfect hexagonal pattern in the Fourier transform (Fig. 2a, b). A relatively strong defocus was applied to obtain sufficient contrast. Indeed, these edge areas are substantially more volatile under beam irradiation (electron flux density < 20 e Å$^{-2}$ s$^{-1}$) compared to large single-crystalline sections. The surface of this edge area is near-atomically smooth along the $\langle \bar{1}100 \rangle$ direction, which suggests an exposed $\{\bar{2}110\}$ surface assuming a vertical facet.

To analyze the spatial distribution of the nanoscale defects, we developed a lattice amplitude mapping approach to evaluate the local crystal misorientation from the zone axis (see Methods). Colors in the map represent the relative strength of the lattice patterns in the three $\{\bar{1}100\}$ directions. In this map (Fig. 2c), a white color suggests that the local area's zone axis [0001] is perfectly aligned with the incident electrons. A red color, for example, suggests that the local area is tilted by rotating around axis $A$ in Fig. 2c. In this case, we only focus on the color balance rather than pixel intensities, as the varying thickness and the oscillatory contrast transfer function may complicate the latter. Nonetheless, nanostructures identified from lattice amplitude maps are generally insensitive to defocus, and only minor shifts in the overall color balance were observed at different defocus settings due to the slight imbalance of the beam deflectors (Supplementary Fig. 12). The lattice amplitude map at the defective edge reveals many subdomains on the 10- to 20-nm scale that are tilted away from each other despite the appearance of single-crystallinity according to the diffraction pattern (Fig. 1c) and the Fourier transform (Fig. 2b). This observation contrasts with that of the highly continuous lattice, indicative of high-quality single crystals with minimal nanoscale defects, away from the thin edges (Supplementary Fig. 12).

By converting the red-green-blue (RGB) model of the composite image into the hue-saturation-intensity (HSI) model, we can further evaluate the orientation and the relative degree of the tilt (Fig. 2e, f). In this model, the pixel saturation represents the color balance, therefore is directly correlated to the orientation which the local crystal domain tilts toward. Here, the subdomain orientation shows a significantly wider distribution compared to that in continuous lattices

(Supplementary Fig. 13). The two dominant orientations are ≈9° and ≈127°, suggesting that the subdomain tilting towards the $a$ axes (rotating around the normal vector of a primary prism plane) is slightly preferred. More importantly, the pixel saturation is associated with the relative tilt of the $c$ axis from the imaging direction. This is a semi-quantitative evaluation because the absolute value depends on the crystal thickness. The saturation histogram reveals that the tilt angle is distributed over a wide range with multiple broad peaks, indicating the presence of shallow local energy minima.

The HRTEM image further reveals the interfacial structures (Fig. 2d). Here, neighboring subdomains connect through relatively sharp interfaces or show gradual distortion over a few nanometers (i.e., mild change in the anisotropy of the lattice patterns). The typical tilt angle of such structures is on the scale of 1° to allow for the visibility of the ice $\{\bar{1}100\}$ patterns according to both lattice visibility criteria[43,44] (Supplementary Fig. 17) and kinematical TEM simulations (Supplementary Figs. 18–29). Notably, a wide variety of interfaces exist with different tilt angles, orientations, and length scales.

MD simulations were employed to study the interfacial structure and energy landscape of these configurations. We used a coarse-grained machine-learned bond order potential (ML-BOP) model of water, which correctly captured the thermodynamic properties of water phases in good agreement with experiments[23]. A low-angle grain boundary (LAGB) between two single-crystalline domains was constructed by rotating one around the $a_2$ axis, annealing at 260 K, and cooling to 93 K (see "Methods" and Supplementary Note 2). We systematically varied the sample thickness and initial tilt angle to evaluate the post-annealing structures and energetics in a broad structural space (see Supplementary Data 1).

For very thin, freestanding ice structures consisting of a few molecular layers (thickness: ≈4.4 nm; $h \leq 2$, where $h$ is the number of supercells (3 unit cells in the $c$ axis, ≈22 Å)), the post-annealing tilt angle significantly deviates from the initial setup (Fig. 3a, upper panel). A wide range of tilt angles are unstable as structures bounce back or forward during annealing. As ice thickness increases, the range of unstable configurations and the tilt angle deviation during annealing are reduced. The same trend is also reflected in the energy landscape of the annealed structures (Fig. 3a, lower panel). A substantial energy variance was observed for few-layer structures, indicating their instability. However, for thicker films of $h = 6$ and 12 (thickness: ≈13.2 and ≈26.3 nm), the energy profile converges and flattens out for tilt angles over ≈0.5° after an initial increase. This observation suggests that the energy penalties for varying the tilt angle of LAGBs in ice films may be minuscule.

To further understand the structural details of these boundaries, we examined the post-annealing MD trajectories ($h = 6$; Fig. 3b). With a low tilt angle (case 1, 0.34°), a defective region with disordered molecules and complex dislocations forms because the low tilt angle is insufficient to support a low-energy dislocation. This result corresponds to the initial energy rise as the tilt angle increases (Fig. 3a, lower panel). When the tilt angle is sufficiently large (case 3, 1.70°), a perfect edge dislocation (Burgers vector, $\mathbf{b} = \frac{1}{3}\langle \bar{1}2\bar{1}0 \rangle$) forms to compensate for the lattice mismatch (Fig. 3b). Interestingly, a moderate tilt angle (case 2, 0.96°) leads to a mixed edge and screw dislocation, with the Burgers vector roughly pointing to the $a_3$ axis. If we look along the $a_2$ axis, i.e., roughly along the dislocation line (Fig. 3c), the crystal in case 2 only has a sufficient tilt angle to accommodate a visual half-plane mismatch, in contrast to the one-unit-cell mismatch in case 3. To avoid the unstable partial edge dislocation, a partial screw dislocation was also generated, leading to a perfect unit cell mismatch (Fig. 3d and Supplementary Fig. 34). This behavior is repeatedly seen in crystals with a larger thickness and when more than one dislocation line is present (Supplementary Fig. 35).

Based on the analyses above, we summarize how ice films develop LAGBs while minimizing the energy penalty (Fig. 3e). In

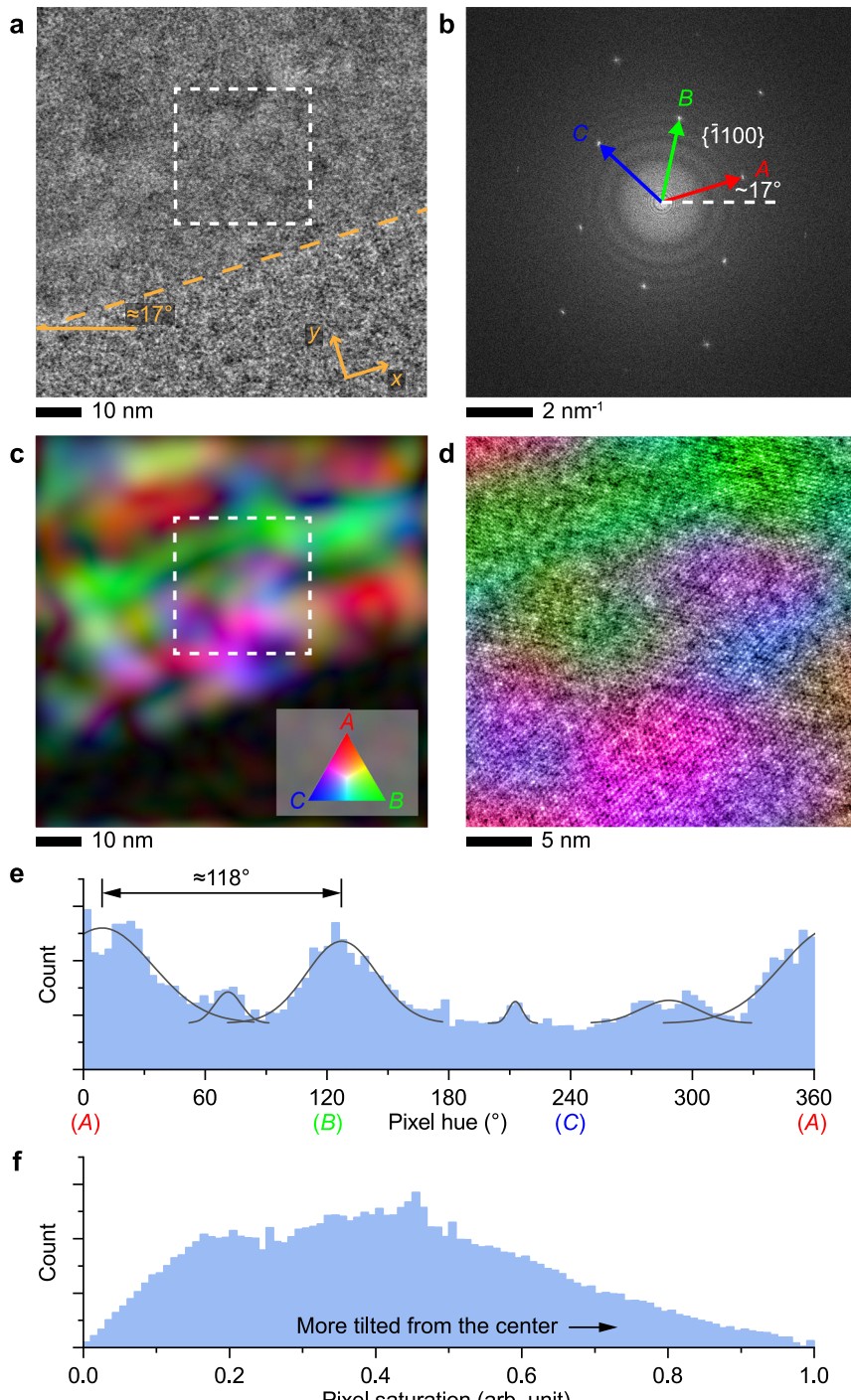

**Fig. 2 | Nanoscale subdomain configuration at a defective crystal edge.**
**a** HRTEM image of a defective ice section near the crystal edge (orange dashed line is a visual guideline). **b** Fourier transform of image **a** with three pairs of {1̄100} reflections labeled as *A*, *B*, and *C*. **c** Intensity map of the three reflections in **b** indexed by red, green, and blue colors. **d** ABS-filtered HRTEM from the area highlighted by the dashed box in **a** colored by (**c**). Histograms of pixel hue (**e**) and saturation (**f**) in the ice area in (**c**) in the hue-saturation-intensity (HSI) model. Curves in the upper panel are Gaussian fitting results. Source data are provided as a Source Data file.

conventional inorganic materials, such as bulk metals (upper schematic), the LAGB energy increases monotonously against the tilt angle with increasing dislocation density[45]. In comparison, ice crystals are mediated by the weak hydrogen bonds and organized by the flexible tetrahedral structural motifs of water molecules[46]. This feature enables a high tolerance to local lattice distortions, including disordered structures and dislocations, along with interfacial configurations characteristic of clathrates (e.g., interfacial beads

identified in Fig. 3). A large tilt angle gap exists in few-layer free-standing ice structures due to the unstable configuration (middle schematic). In ice films thicker than ≈10 nm relevant to our TEM experiments (lower schematic), the energy of the LAGB first increases due to elastic strain and the formation of disordered structures but then quickly flattens out over a wide range of tilt angles, owing to the formation of perfect dislocations regardless of the tilt angle. The presence of unstable configurations and shallow local energy minima

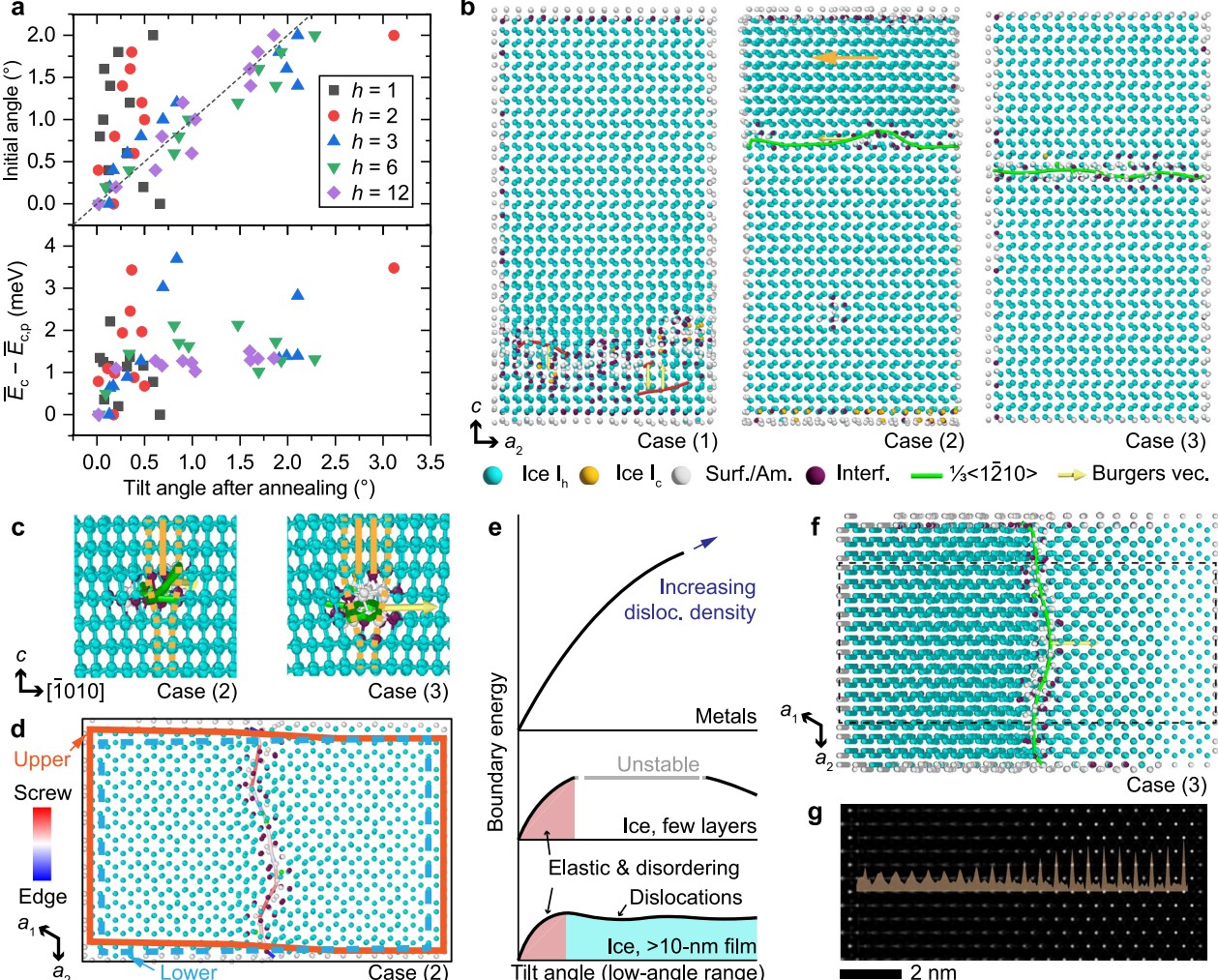

**Fig. 3 | Molecular dynamics studies of low-angle grain boundaries in ice films.** **a** The initial angle (upper panel) and the mean cohesive energy per molecule of annealed MD models at 93 K ($\bar{E}_c$) compared to that of an annealed perfect crystal ($\bar{E}_{c,p}$; lower panel) as a function of the final tilt angle. Crystal thickness ($h$) is expressed in the number of supercells (3 unit cells in the $c$ axis, ≈22 Å). **b** Cross-sectional models of the tilted ice ($h = 6$) with a final tilt angle of 0.34° (1), 0.96° (2), and 1.70° (3). An orange arrow indicates domain distortion. **c** Cross-sectional models of the dislocation core of cases (2) and (3). Orange lines and dashes represent atomic planes around the dislocation. **d** Cross-sectional model of case (2) showing molecules on the slip plane, the configuration of upper and lower domains, and the local character of the dislocation. **e** Schematic energy diagrams of low-angle grain boundaries in typical metals, freestanding few-layer ice, and thicker ice films >10 nm. Top-down view of the case (3) with cross-section exposing the dislocation core (**f**) and multislice-simulated TEM image of the crystal in the dashed-box area (**g**). Defocus: −55 nm. The brown trace in (**g**) is the intensity profile of a row of lattice patterns. All structural models share a legend below (**b**). Beads: water molecules in ice, surface/amorphous (surf./am.), interfacial (interf.), or hydrate-like local configurations (other colors). Red curves represent dislocations of other types. Source data are provided as a Source Data file.

could further explain the mild preference for certain tilt angles observed in the experiment (Fig. 2e, lower panel). As such, LAGBs with various tilt angles can co-exist in ice films with low energy penalties.

This conclusion is consistent with our TEM results. The annealed configuration from the MD (case 3) was used for multislice TEM simulation (Fig. 3f, g). The simulated TEM image correctly reproduced the isotropic hexagonal lattices on the right that are perfectly aligned with the [0001] zone axis. Meanwhile, the pattern on the left is elongated in the horizontal direction due to tilting, consistent with the lattice anisotropy observed from experimental images in tilted subdomains (Fig. 2d). The increase in anisotropy is also evidenced by the horizontal intensity profile for a row of lattice spots in Fig. 3g. Importantly, even though a single dislocation line is present in the middle, the pattern anisotropy exhibits a gradual transition across 1 to 2 nm, consistent with the interfacial transition observed in the experiments.

## Trapped gas bubbles

Away from the defective edges described above, we investigated the interior regions of single-crystalline ice. Continuous hexagonal lattices were observed over large areas (Fig. 4a–c), and the lattice amplitude map reveals an absence of subdomains (Fig. 4d).

We discovered many trapped gas bubbles in the form of nanoscale cavities in the crystal (Fig. 4a, circular and elliptical features labeled by arrows). Thickness gradients due to bubble curvature are evidenced by the significant contrast variations near the bubble surfaces. The formation of nanosized trapped bubbles is attributed to the relatively fast cooling rate of the thin sample compared to bulk freezing[47]. Note that these bubbles exist prior to imaging and remain unchanged throughout the imaging process. Therefore, it is most likely that they are generated by gas precipitation instead of radiolysis.

By calculating the in-plane lattice distortion tensors using geometric phase analysis (GPA)[48], we found that almost no additional

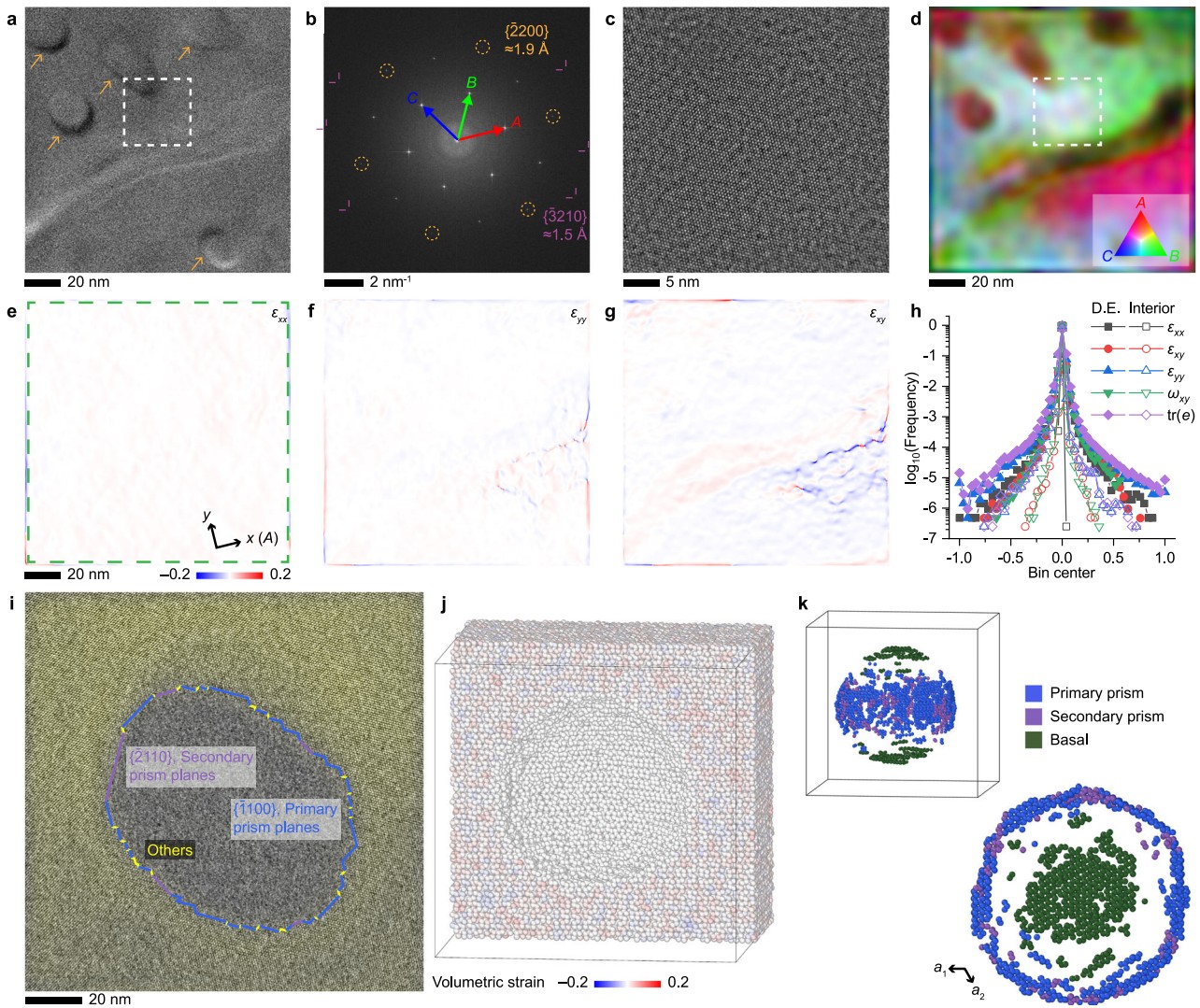

**Fig. 4 | Trapped gas nanobubbles formed by water crystallization. a** HRTEM image of an ice $I_h$ crystal aligned to the [0001] zone axis from the interior area of a crystal section. Arrows indicate trapped gas bubbles. **b** Fourier transform of (**a**). **c** ABS-filtered HRTEM image from the area highlighted by the dashed box in (**a**). **d** Intensity map of the three reflections in **b** indexed by red, green, and blue colors. Maps of in-plane strain distribution from GPA for $xx$ (**e**), $yy$ (**f**), and $xy$ (**g**). **h** Histogram of mechanical quantities in this area (Interior) and the defective edge

(D.E.; Fig. 2a). **i** ABS-filtered HRTEM image of a through-hole in thin ice films. The exposed crystal plane assignment assumes vertical facets with a projected length of at least 3 unit cells. A yellow shade indicates areas showing ice lattices. **j** Cross-sectional view of an MD-simulated nanobubble ($r = 6$ nm) in ice color-coded by the local volumetric strain. **k** Surface beads from the nanobubble color-coded by recognized facets in 3D and top-down views. Source data are provided as a Source Data file.

strain field exists in the crystals surrounding the nanobubbles (Fig. 4e–g and Supplementary Fig. 16). The distribution of all in-plane lattice distortion indicators (strains, rotation, and dilation) is significantly narrower than that of the subdomain-containing defective edge (Fig. 4h).

Previous MD simulations for metals, such as Al, predicted strain fields surrounding a nanocavity ($r = 10$ nm) on the scale of 1–3%[49]. To understand the absence of such a strain field in ice, we first used continuum theories of elasticity to estimate the strain distribution around a spherical cavity[50]. The maximum elastic strains caused by the Laplace pressure of an ice nanobubble ($r = 10$ nm) is <0.4% (Supplementary Fig. 35). As such, nanobubbles in ice should be expected to cause negligible strain fields, which agrees with the HRTEM-derived strain maps.

Our MD simulations also confirm this conclusion. We modeled a spherical cavity ($r = 6$ nm) in an ice single crystal, melted the structures within 3 nm in the vicinity of the cavity surface, and recrystallized them in silico (see Methods and Supplementary Data 2). Volumetric strain

analysis shows no discernable strain accumulation around the cavity, with a mean and standard deviation of only 0.30% and 1.00% (Fig. 4j).

To further investigate the inner surface structure of the bubbles on the molecular scale, we moved to single-crystalline ice regions with a much lower thickness, where bubbles became quasi-cylindrical through-holes (Fig. 4i and Supplementary Movie 3). The reduced thickness allowed direct imaging and recognition of the lattice structures near the bubble surface. Notably, despite the overall curved shape, most exposed surfaces on the molecular scale can be attributed either to the primary prism planes, $\{\bar{1}100\}$, or secondary prism planes, $\{\bar{2}110\}$, assuming vertical facets. These two surfaces are the lowest-energy facets in ice $I_h$ perpendicular to [0001][51,52]. The ratio between the projected length of the primary and secondary prism planes is circa 3.8:1 (Table 1).

The bubble model in MD simulations was also used to reveal the surface faceting at the molecular level (see Supplementary Note 2.2). A significant fraction of the surface molecules are exposed as basal, primary prism, or secondary prism planes (Fig. 4k and Table 1). The

**Table 1 | Molecular-scale facet composition on nanobubble surfaces**

| Method | Basal | Primary prism | Secondary prism | Other | PP/SP ratio[a] |
|---|---|---|---|---|---|
| HRTEM (2D) | – | 68.8% | 17.9% | 13.3% | 3.84 |
| MD (3D) | 8.21% | 18.5% | 4.68% | 68.6% | 3.95 |
| MD cross-section[b] | – | 40.8% | 10.3% | 48.9% | 3.96 |
| Energy (meV Å$^{-2}$)[c] | 6.48 | 6.78 | 7.82 | – | – |

Percentages represent length fractions measured from HRTEM along [0001] and molecular coverages from MD.

[a]Ratio between primary and secondary prism planes.

[b]Cross-section (thickness = 5 nm) of the bubble center perpendicular to [0001].

[c]Excess free surface energy of the facet at 0 K calculated from MD.

fractional ratio between the primary and secondary prism planes in the MD-simulated cross-section at the center (Supplementary Fig. 38) and HRTEM follow the same trend predicted by their surface energy (Table 1). This result indicates that the trapped gas bubbles in the experiments are very close to thermodynamic equilibrium, and the nanoscale configurations we observed could reflect those in sufficiently annealed bubbles. We note that a lower amount of basal planes were observed in MD than prism planes despite their lower surface energy because the former must fit into the top and bottom curvatures. As such, trapped bubbles in ice are defined both by the macroscopic rounded shape and molecular-scale facets to minimize the total surface energy. The presence of these nanofacets may be important in defining the stability and migration kinetics of trapped gas bubbles in glacial systems[16,53].

### In-situ observation of dynamic bubble trajectories

When we elevated the stage temperature to −70 °C and reduced the electron flux density to 25 e Å$^{-2}$ s$^{-1}$, we observed nucleation and growth of new nanobubbles in single-crystalline ice sections under HRTEM conditions (Fig. 5a and Supplementary Movie 4). The newly generated bubbles show similar shapes with trapped nanobubbles imaged at about −180 °C. Furthermore, these bubbles can migrate in the crystal and completely dissolve under the same conditions (Fig. 5b and Supplementary Movie 5) while the ice sample stays single-crystalline, suggesting that the system is near a steady state for bubble generation and ice recrystallization.

Bubbles can also dynamically coalesce and merge into larger ones (Fig. 5c and Supplementary Movie 6). In this observation, two bubbles approached each other, and their outlines initially showed different thickness contrasts. However, the contrast matched upon coalescence, which suggests that the two bubbles physically connected rather than passed by at different vertical positions. The ice surfaces in the bubbles are highly dynamic, as shown by the significant reshaping of the merging ones.

The persistent hexagonal patterns in the Fourier transform of the movies suggest that the ice sample stayed single-crystalline in these observations. GPA was performed on all frames to evaluate the mechanical consequences of the bubble trajectories. Here, the mean value of in-plane lattice dilation reflects the global distortion of the sample and only fluctuated within a range of -1% throughout the experiments (Fig. 5d). The standard deviation of the lattice dilation was further calculated to evaluate local lattice distortion (Fig. 5e). Indeed, most data fall between the levels observed in the defective edges (Fig. 2) and the bubble-containing interior sections (Fig. 4) at about −180 °C. These observations further show that nanobubbles cause insignificant strain fields in ice crystals even during their dynamic evolution.

Bubble formation in ice under the electron beam can mainly be attributed to radiolysis[54] and knock-on damage by the incident electrons[55]. To evaluate the radiolytic chemistry under the experimental conditions, we extended approaches developed for water radiolysis in liquid-phase electron microscopy[56,57] to lower temperatures and numerically calculated the reaction kinetics at −70 °C and 25 e Å$^{-2}$ s$^{-1}$ (see Methods and Supplementary Data 3). Notably, the system reaches a steady state within seconds of gas generation (Fig. 5f), consistent with in-situ TEM results. It is crucial to recognize that radiolysis offers one credible explanation for the formation of new bubbles in these experiments, but detecting the content of the gas species is challenging in these experiments. Nonetheless, the formation of substantial $H_2$ and $O_2$ species is in agreement with cryogenic EELS analyses of amorphous ice[58,59], highlighting the potential role of these gas molecules in constituting bubble volumes in our experiments.

## Discussion

We have presented the first molecular-resolution imaging of nanoscopic defects in ice $I_h$ crystallized from liquid water. The presented sample preparation method, which crystallized liquid water into thin ice films between a-C membranes, was a critical factor that led to this imaging breakthrough. These samples contain large-area single-crystalline regions that are sufficiently stable under HRTEM conditions for extended imaging. This method could be extended to prepare samples of mixed aqueous systems, as well as organic solvents, and provide new insights into liquid-liquid phase separation, solute precipitation, and the freezing behavior of organic liquids. Despite the presence of a heterogeneous interface between ice/water and the encapsulating membrane, the ice sample thickness is typically well beyond the thickness of interfacial regions (on the order of 1 nm)[60] such that there is no significant deviation from the physics of bulk phase transformation. However, the interfacial equilibrium and properties at this length scale are anticipated to differ from those of the bulk. Therefore, caution must be exercised when translating conclusions from HRTEM studies to bulk crystals. By maneuvering the sample temperature and electron flux density, a near-steady state of bubble generation and dissolution in crystalline ice was achieved. A finer control over the imaging conditions will likely pave the way to the direct imaging of ice-water interfacial structures and, moreover, crystallization and melting dynamics with molecular resolution, a holy grail in the ice research community[61–63]. Experimental images and movies can now be directly compared and correlated to computations on an previously unattainable molecular length scale to reveal the underlying structures, molecular interactions, and phase transformation pathways. The direct extraction of such Å-scale information provides a new research paradigm for theory, modeling, and forecasting of ice crystallization and melting in environmental, biological, and material systems.

## Methods

### Materials

Deionized (DI) water was generated by an ELGA PURELAB flex 2 system with a resistivity of 18.2 MΩ cm. Carbon-coated Cu grids for TEM were purchased from Ted Pella, Inc. Ammonium persulfate (≥98%) was purchased from Sigma-Aldrich.

### Preparation of encapsulated ice $I_h$ samples

In a typical experiment, 1–3 μL of DI water was dispensed on the carbon side of a TEM grid and loaded on a Gatan 626 single-tilt or a 915 double-tilt liquid nitrogen cryo-transfer holder, with the carbon/water side facing up. Another identical grid was put on top of the first one, with the carbon side facing down. The grid orientation was roughly aligned when placing the second grid, and the surface tension of water would then align and adhere the grids together. Depending on the models of the TEM and their column vacuum level, the sample can either be frozen inside or outside the column with liquid nitrogen. Freezing outside the TEM in the cryo transfer station is typically preferred for

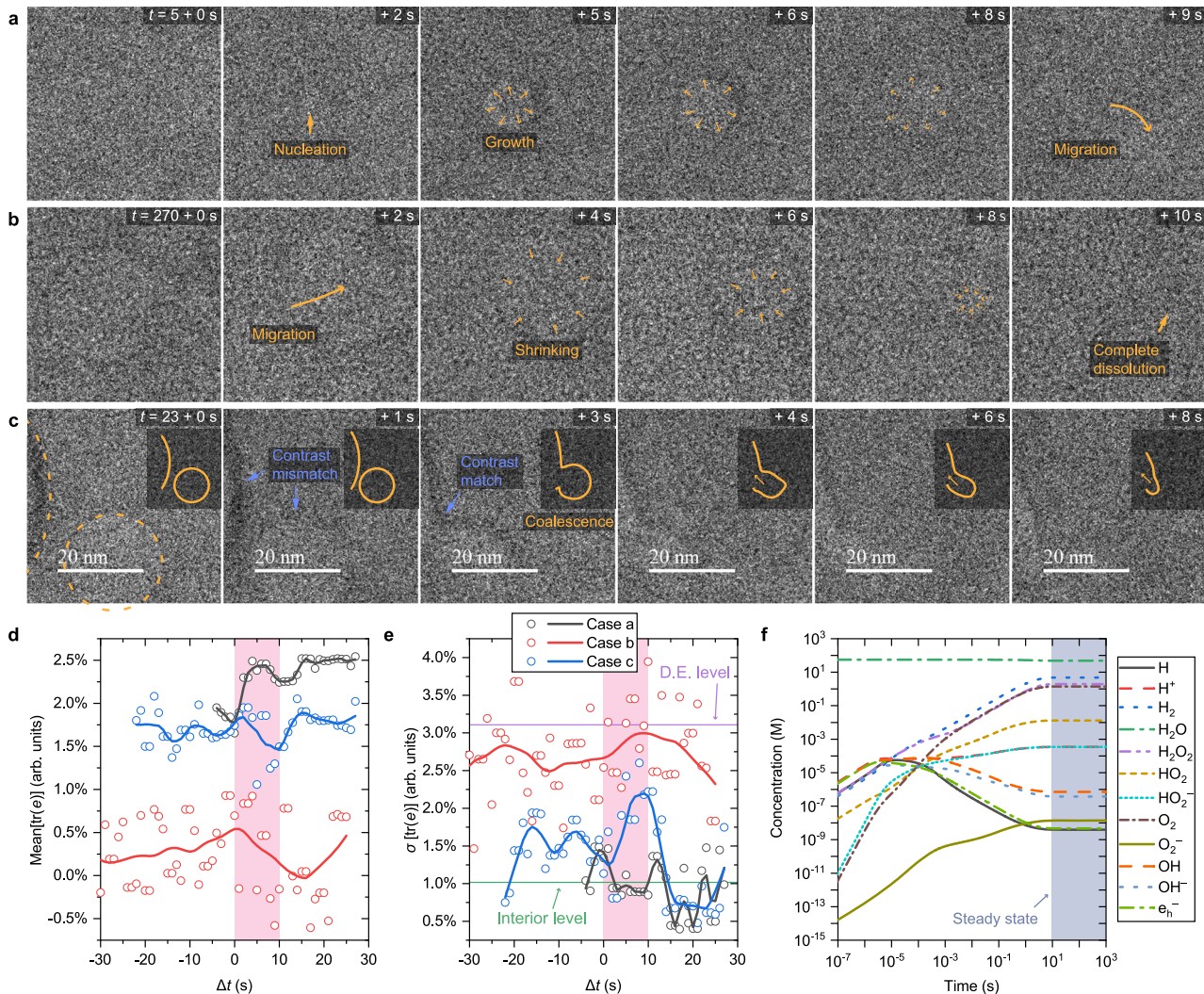

**Fig. 5 | Direct observation of gas bubble trajectories in single-crystalline ice I$_h$ near a steady state.** Time-sequence drift-corrected HRTEM images at a stage temperature of −70 °C and an electron flux density of 25 e Å$^{-2}$ s$^{-1}$ showing three types of bubble dynamics: nucleation and growth (**a**), dissolution (**b**), and coalescence (**c**). Insets in (**c**): solid curves represent the shape evolution of the bubble outlines (dashed curves in the first panel). All images share the scale bars shown in (**c**). The mean (**d**) and standard deviation (**e**) of lattice dilation [tr(*e*)] as a function of time from three HRTEM sequences [**a**–**c**; Supplementary Movies 4–6]. Pink shades indicate the period shown in (**a**–**c**). Curves are smoothing results (LOWESS, span = 0.15). D.E. defective edge. **f** Calculated ice radiolysis kinetics at −70 °C and an electron flux density of 25 e Å$^{-2}$ s$^{-1}$. Source data are provided as a Source Data file.

consistency. In either case, the copper shield remains closed during sample freezing.

## Cryo-TEM and data processing

TEM was performed on an FEI Titan Environmental TEM [extreme-brightness Schottky field-emission gun (X-FEG), 300 kV] equipped with a CEOS double-hexapole aberration corrector (CETCOR) for the image-forming lenses and a Gatan UltraScan 1000 charge-coupled device (CCD) scintillation camera. A Gatan Metro 300 direct electron detector was used for the sample stability studies. TEM along zone axes other than [0001] was performed on an FEI Themis TEM (X-FEG, 300 kV) equipped with a CEOS CETCOR for the image-forming lenses and an FEI Ceta 16 M complementary metal-oxide-semiconductor (CMOS) scintillation camera. Low-loss EELS was performed on a JEOL JEM-ARM300CF GRAND ARM (cold FEG, 300 kV) equipped with a JEOL dodecapole expanding trajectory aberration corrector for the probe-forming lenses and a Gatan Imaging Filter (GIF) Quantum system. Convergence angle: 41.2 mrad; EELS collection angle: 62.4 mrad. EELS thickness measurements were conducted on an FEI Themis equipped with a GIF Quantum system with a convergence angle of 17.9 mrad and

a collection angle of 36.18 mrad. Energy-filtered TEM was done on the same system. During the experiments, the temperature readout from the cryo holder is typically between −180 and −178 °C.

For all Fourier transform images presented in this article, the modulus of the complex values are displayed and are scaled linearly. Average background subtraction (ABS) filtering[64] of high-resolution TEM (HRTEM) images was performed using a script in Gatan DigitalMicrograph (http://www.dmscripting.com/hrtem_filter.html) with the following parameters: Delta = 5.0%, BW *n* = 4, BW Ro = 0.8, and low-frequency tapering = 1.0%. Low-loss EELS profiles were first processed by removing the plural scattering using the Fourier-log algorithm. All EELS profiles were smoothed by adjacent averaging with a 50-pixel moving window size.

## Calculation of lattice maps from electron images

The lattice amplitude maps were calculated with the following procedures: (1) obtain the Fourier transform image; (2) mask only the circular areas containing the desired pair of lattice reflections and assign zero to all other pixels (mask diameter = 0.2 nm$^{-1}$); (3) perform inverse Fourier transformation; (4) obtain the absolute value of the

image (i.e., flip the sign of negative intensity values); (5) perform Gaussian smoothing with $\sigma = 5$ pixels (this may vary depending on the pixel size); (6) repeat this process for the other two pair of reflections; (7) overlay the three resulting images as a red/green/blue (RGB) stack. The lattice distortion maps (strain, rotation, and dilation) were calculated using the Strain++ package (https://jjppeters.github.io/Strainpp/) based on the geometric phase analysis (GPA) algorithm[48]. Zero-distortion points in GPA were chosen at the maximum-modulus frequencies in the Fourier transform. The Gaussian mask size ($3\sigma$) is set to 1/8 of the $\{\bar{1}100\}$ $g$ vector length.

## In-situ cryogenic electron microscopy
After obtaining an ice sample in TEM, the holder temperature was raised to −70 °C and stabilized. Movies were recorded by screen capturing at an electron flux density of 25 e Å$^{-2}$ s$^{-1}$ determined on the fluorescence screen through the integrated sensor. Frames were registered using the ImageJ StackReg module (http://bigwww.epfl.ch/thevenaz/stackreg/).

## Characterization of the carbon membranes
The thickness and surface roughness of the carbon membranes from the TEM grids were measured based on method described in ref. [65]. Briefly, the copper grid was etched by an ammonium persulfate aqueous solution ($\approx$100 mg mL$^{-1}$), leaving the carbon membrane floating on the solution surface. The carbon membrane was then transferred to a glass slide and then to DI water for cleaning, which was repeated two times. Finally, the membrane was transferred to a silicon wafer ready for atomic force microscopy (AFM) imaging. AFM was performed on an Asylum Research Cypher ES with a Bruker RFESPA-75 probe (resonance frequency, 75 kHz; spring constant, 3 N m$^{-1}$). The scan rate was 1.5 Hz, and the amplitude was between 300 and 350 mV. The thickness was determined at the edge of the membrane ruptures caused by drying. Image processing and quantification were performed with the Gwyddion package[66] (http://gwyddion.net/).

The chemical characteristics of the carbon membrane surface were studied by X-ray photoelectron spectroscopy (XPS) on a Kratos Axis Ultra that uses a monochromatic focused Al $K_\alpha$ X-ray (1486.7 eV) source. The instrument was calibrated prior to the introduction of the sample and referenced to binding energies for Cu $2p^{3/2}$ at 932.67 ± 0.05 eV and Au $4f$ at 84.0 ± 0.05 eV. The base pressure of the sample analysis chamber was maintained at $1.6 \times 10^{-9}$ Torr. The sample was prepared by adhering the carbon-coated copper grid to a stainless-steel bar using copper tape. The tape was used only in the corners, while at least two data points were analyzed from the center of the grid. Since the carbon-coated copper grid was conductive, no neutralizer was applied, and the results were analyzed without any charge correction.

Furthermore, the water contact angle was measured by drop-casting $\approx$1-μL of DI water onto a TEM grid and immediately capturing a side-view image. The drop shape was fitted to calculate the contact angle[67]. Experiments were repeated on three separate TEM grids.

## Simulation of electron microscopy and diffraction
Kinematical HRTEM simulation from single crystals was performed using the ReciPro package[68] (https://seto77.github.io/ReciPro/). The simulation assumes a parallel beam condition, a beam energy of 300 keV ± 0.8 eV, $C_s = 4$ μm, and $C_c = 1.4$ mm. Selected area electron diffraction (SAED) patterns were simulated based on the dynamical theory using ReciPro, assuming a sample thickness of 30 nm. Polycrystal SAED radial profiles were simulated based on the kinematical theory using the CrystalDiffract package (CrystalMaker Software, https://crystalmaker.com/crystaldiffract/). These simulations used the experimentally derived ice I$_h$ crystal structure in the space group $P6_3/mmc$[69,70].

Multislice TEM simulation[71] based on explicit models from coarse-grained MD was performed with the QSTEM package[72] (https://www.physik.hu-berlin.de/en/sem/software/software_qstem) using the same TEM parameters. An F atom was used in place of a water molecule in the coarse-grained MD model for image simulation.

## Coarse-grained molecular dynamics simulations
The MD simulations were performed using a coarse-grained machine-learned bond order potential (ML-BOP) model of water[23]. For grain boundary simulations, minimal-energy configurations of bi-grain ice models with different thicknesses and tilt angles were first established and equilibrated with the conjugate gradient algorithm[73]. The models were subsequently annealed in silico using the LAMMPS package[74] at 260 K and then gradually cooled to 93 K in the microcanonical (NVE) ensemble. For nanobubble simulations, a cavity of a radius of 6 nm was created in a single-crystalline ice I$_h$ model by removing the molecules within. The model was first equilibrated in LAMMPS at 260 K and 1 bar. The surface areas of the bubble (thickness: 3 nm) were subsequently heated to 270 K and then 370 K for melting, and finally cooled back to 260 K for recrystallization. For detailed procedures, see Supplementary Note 2.

The OVITO package[75] (https://www.ovito.org/) was used to perform local symmetry identification using the CHILL+ algorithm[76] and dislocation analysis using an extended dislocation extraction algorithm (DXA)[77].

## Numerical calculation of radiolysis kinetics
Radiation chemistry is modeled by considering the interplay of twelve reactants relevant for liquid-phase TEM (H$_2$O, H$_2$, O$_2$, H$_2$O$_2$, H, OH, HO$_2$, H$^+$, HO$_2^-$, O$_2^-$, OH$^-$, and the solvated electron e$_h^-$). An amorphous ice sample with no spatial anisotropy is considered. The reactions among these reactants span a kinetic model that is described by a set of coupled differential equations[78]:

$$\frac{\partial c_i}{\partial t} = \sum_j k_j \left( \prod_l c_l \right) - \sum_{m \neq j} k_m \left( \prod_n c_n \right) + \rho \psi G_i. \qquad (1)$$

Here, $c_i$ denotes the concentration of the reactant $i$, $t$ the time, $k$ the reaction-rate constant of the respective reaction, $G_i$ the generation value (G-value) of $i$ upon electron irradiation, and $\rho$ the density of amorphous ice (0.92 g L$^{-1}$)[79]. The dose rate $\psi$ is calculated from Eq. (2)[80]:

$$\psi = \frac{S}{e} \phi. \qquad (2)$$

In Eq. (2), $\phi$ is the electron flux density, and $e$ is the elementary charge. As the inelastic scattering of 300 keV electrons in amorphous ice appears to be reasonably close to that in water[81,82], a density-normalized stopping power $S$ of 2.36 MeV cm$^{-2}$ g$^{-1}$ was approximated[83]. For an electron-flux density of 25 e Å$^{-2}$ s$^{-1}$, this yields a dose rate of $9.45 \times 10^7$ Gy s$^{-1}$.

The harnessed kinetic model extrapolates Arrhenius-based rate constants[84] to lower temperatures (Supplementary Table 8). As cryogenic G values are unavailable[59], room temperature values are used (Supplementary Table 9)[85]. Modeling is performed using a temperature-dependent extension of AuRaCh[57]. The code is available at https://github.com/BirkFritsch/Radiolysis-simulations.

## Data availability
The data that support the findings of this study are available from the corresponding author upon request. MD simulation models are provided as Supplementary Data 1, 2 and the radiolysis calculation data is provided as Supplementary Data 3. Source data are provided with this paper.

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

## Acknowledgements

We thank L. Kovarik, D. Li, and M. Zhang of Pacific Northwest National Laboratory (PNNL) and K. Bustillo, R. Dhall, and J. Ciston of Lawrence Berkeley National Laboratory (LBNL) for their helpful discussions. We thank J.J.P. Peters (Trinity College Dublin) for implementing stack processing in the Strain++ package. Microscopy and analysis were supported by the U.S. Department of Energy (DOE) Office of Science (SC) Basic Energy Sciences (BES) Division of Materials Science and Engineering, Synthesis and Processing Sciences program (FWP 67554) at PNNL (J.J.D.Y.). Molecular dynamics simulations were supported by the Data, Artificial Intelligence, and Machine Learning at Scientific User Facilities program under the Digital Twin Project at Argonne National Laboratory (S.K.R.S.S.). Development of ice encapsulation and imaging methodology was supported by a U.S. DOE SC Distinguished Scientist Fellows award (FWP 77246) at PNNL (J.J.D.Y.). A portion of this research was performed on project awards (60286, 60620, and 60789) from the Environmental Molecular Sciences Laboratory at PNNL (J.S.D. and J.J.D.Y.). Work at the Molecular Foundry and the National Energy Research Scientific Computing Center was supported by the U.S. DOE SC BES under Contract No. DE-AC02-05CH11231. Work at the Center for Nanoscale Materials was supported by the U.S. DOE SC BES under Contract No. DE-AC02-06CH11357. J.S.D. acknowledges a Washington Research Foundation Postdoctoral Fellowship. PNNL is a multiprogram national laboratory operated for the DOE by Battelle under Contract DE-AC05-76RL01830.

## Author contributions

J.S.D. and J.J.D.Y. conceived and designed the study. J.S.D. performed electron microscopy and its simulations. H.C. and S.K.R.S.S. developed the machine-learned coarse-grained model for water. S.B. and H.C. performed molecular dynamics simulations. B.F. and A.H. performed radiolysis calculations. Y.X., A.S.K. and J.S.D. characterized the membrane surface. J.J.D.Y., S.K.R.S.S. and A.H. supervised the study. J.S.D. led manuscript drafting. All authors analyzed and discussed the data and revised the manuscript.

## Competing interests

The authors declare no competing interests.
