## [Transparent Peer Review file · Nature Communications]

Molecular-Resolution Imaging of Ice Crystallized from Liquid Water by Cryogenic Liquid-Cell TEM

Corresponding Author: Dr James De Yoreo

Version 0:

Reviewer comments:

Reviewer #1

(Remarks to the Author)

The manuscript titled "Molecular-Resolution Imaging of Ice Crystallized from Liquid Water" by Du et. al. introduces a method for stabilizing ice, specifically ice I_h directly crystallized from liquid water. Conventional TEM sample preparation techniques for ice rely on a plunge freezing method to rapidly freeze a very thin layer of water kept at high humidity, which produces a thin layer of ice but is flash frozen at very low temperatures (near ~ -160 C), producing an amorphous form of ice. Directly freezing a thin layer of water into an equilibrium ice I_h phase is normally difficult, particularly because water films tend to evaporate before freezing into such phases. The authors resolve this by using a carbon-encapsulated cryo liquid cell to prevent water from drying while the ice is freezing between the carbon films.

This form of imaging is indeed suitable to reveal structures of ice at very high resolutions, which reaches scales that are typically inaccessible with spectroscopy or diffraction techniques. It is also worth mentioning that the method inherently focuses on observations of ice that are very thin, which brings novel insight into the different structures of ice that are affected by interfacial tension. The results of the imaging of thin ice reveal the presence of tilted orientations of crystalline domains at ice edges, while also showing that strain fields are not present at bubble interfaces from those trapped within ice. The reviewer believes that the manuscript is suitable for publication in Nature Communications, but has some revisions that should be addressed.

1. The observation technique mainly focuses on ice observed at very thin scales, which could possess different properties from the bulk. I think the authors should include this discussion in the manuscript.
2. Considering that a novel methodology to preserve liquid water while it is freezing for preparing TEM samples has been presented, I suggest the authors to discuss potential future uses for this sample preparation method.
3. Could the methodology be used to observe other phases of ice? Particularly those near no-man's land or those that are difficult to observe with other forms of analysis?

Reviewer #2

(Remarks to the Author)

The authors have done additional works in addressing my comments. The revised manuscript is now suitable for publication in Nature Communications.

Reviewer #3

(Remarks to the Author)

This manuscript presents outstanding work by combining cryogenic TEM with molecular dynamics simulations to reveal molecular-scale structures and dynamics in crystalline ice formed from liquid water. The data quality, experimental design, and interpretative details are all outstanding, and the findings, particularly regarding nanobubble behavior and defect tolerance, portray a significant advancement in the field. One particularly unique and commendable aspect is that the

molecular resolution is achieved directly through ice crystallized from liquid water. This sets the work apart and adds value beyond the resolution itself. The integration of in situ imaging with defect and nanobubble analysis is especially insightful, and the paper is written with well-supported conclusions. Aside from the below-mentioned minor issues related to terminology and clarity, there are no major concerns.

1. The manuscript presents compelling evidence for the presence of nanoscale bubbles within single-crystalline ice, observed via cryo-TEM and supported by molecular dynamics simulations and strain analysis. The faceting and negligible surrounding strain fields strongly suggest that these are stable features, plausibly formed during crystallization or beam-induced radiolysis.

That said, I found myself wondering: how certain can we be that these are truly gas-filled bubbles? The interpretation certainly makes sense, especially given the modeling and the known effects of beam-induced radiolysis. But since there's no direct compositional data like EELS taken specifically from the bubble regions, I think it's worth acknowledging that other possibilities, like vacuum voids or sublimation-induced cavities, can't be entirely ruled out in this kind of experiment.

To be clear, I don't see this as a flaw in the study. It's a tough experimental challenge, and the authors already do a great job with what's available. But I'd suggest briefly flagging this point in the main text or discussion. A sentence or two noting the inferential nature of the gas identification and perhaps pointing toward future work that could directly probe bubble content, would add helpful nuance without weakening the main conclusions.

2. In the Main section (line 105), authors refer to "lattice amplitude maps" as a central tool for visualizing and subdomain structure and crystal misorientation. Could the authors consider briefly explaining what a lattice amplitude map represents, how it is constructed and what type of structural information it reveals? A short clarification would help readers better understand who may not be familiar with this analysis technique.

Additionally, would it be possible to distinguish more explicitly how this approach differs from geometric phase analysis (GPA)?

3. In the section discussing MD simulations of low-angle grain boundaries (e.g., Fig. 3), the authors refer to the presence of "clathrate". Could the authors clarify what is meant by this term in the context of the simulations?

Response to the Reviewer Comments

Reviewer #1

The manuscript titled “Molecular-Resolution Imaging of Ice Crystallized from Liquid Water” by Du et. al. introduces a method for stabilizing ice, specifically ice Ih directly crystallized from liquid water. Conventional TEM sample preparation techniques for ice rely on a plunge freezing method to rapidly freeze a very thin layer of water kept at high humidity, which produces a thin layer of ice but is flash frozen at very low temperatures (near ~ -160 C), producing an amorphous form of ice. Directly freezing a thin layer of water into an equilibrium ice Ih phase is normally difficult, particularly because water films tend to evaporate before freezing into such phases. The authors resolve this by using a carbon-encapsulated cryo liquid cell to prevent water from drying while the ice is freezing between the carbon films.

This form of imaging is indeed suitable to reveal structures of ice at very high resolutions, which reaches scales that are typically inaccessible with spectroscopy or diffraction techniques. It is also worth mentioning that the method inherently focuses on observations of ice that are very thin, which brings novel insight into the different structures of ice that are affected by interfacial tension. The results of the imaging of thin ice reveal the presence of tilted orientations of crystalline domains at ice edges, while also showing that strain fields are not present at bubble interfaces from those trapped within ice. The reviewer believes that the manuscript is suitable for publication in Nature Communications, but has some revisions that should be addressed.

1. The observation technique mainly focuses on ice observed at very thin scales, which could possess different properties from the bulk. I think the authors should include this discussion in the manuscript.

Response: We added a discussion on the implication of the sample thickness to the manuscript on page 11, lines 276–278:

Despite the presence of a heterogeneous interface between ice/water and the encapsulating membrane, the ice sample thickness is typically well beyond the thickness of interfacial regions (on the order of 1 nm)⁶⁰ such that there is no significant deviation from the physics of bulk phase transformation. **However, the interfacial equilibrium and properties at this length scale are anticipated to differ from those of the bulk. Therefore, caution must be exercised when translating conclusions from HRTEM studies to bulk crystals.**

2. Considering that a novel methodology to preserve liquid water while it is freezing for preparing TEM samples has been presented, I suggest the authors to discuss potential future uses for this sample preparation method.

Response: A new discussion was added to the manuscript on page 11, lines 271–273:

These samples contain large-area single-crystalline regions that are sufficiently stable under HRTEM conditions for extended imaging. **This method could be extended to prepare samples of mixed aqueous systems, as well as organic solvents, and provide new insights into liquid-liquid phase separation, solute precipitation, and the freezing behavior of organic liquids.**

3. Could the methodology be used to observe other phases of ice? Particularly those near no-man's land or those that are difficult to observe with other forms of analysis?

Response: The samples in this study were prepared at ambient pressure, resulting in the observation of type I ice exclusively. While there may be potential approaches to incorporate high-pressure systems to access other ice phases, such explorations fall beyond the scope of the current study.

Reviewer #2

The authors have done additional works in addressing my comments. The revised manuscript is now suitable for publication in Nature Communications.

Response: Thank you for your contributions to improving this manuscript.

Reviewer #3

This manuscript presents outstanding work by combining cryogenic TEM with molecular dynamics simulations to reveal molecular-scale structures and dynamics in crystalline ice formed from liquid water. The data quality, experimental design, and interpretative details are all outstanding, and the findings, particularly regarding nanobubble behavior and defect tolerance, portray a significant advancement in the field. One particularly unique and commendable aspect is that the molecular resolution is achieved directly through ice crystallized from liquid water. This sets the work apart and adds value beyond the resolution itself. The integration of in situ imaging with defect and nanobubble analysis is especially insightful, and the paper is written with well-supported conclusions. Aside from the below-mentioned minor issues related to terminology and clarity, there are no major concerns.

1. The manuscript presents compelling evidence for the presence of nanoscale bubbles within single-crystalline ice, observed via cryo-TEM and supported by molecular dynamics simulations and strain analysis. The faceting and negligible surrounding strain fields strongly suggest that these are stable features, plausibly formed during crystallization or beam-induced radiolysis.

That said, I found myself wondering: how certain can we be that these are truly gas-filled bubbles? The interpretation certainly makes sense, especially given the modeling and the known effects of beam-induced radiolysis. But since there's no direct compositional data like EELS taken specifically from the bubble regions, I think it's worth acknowledging that other possibilities, like vacuum voids or sublimation-induced cavities, can't be entirely ruled out in this kind of experiment.

To be clear, I don't see this as a flaw in the study. It's a tough experimental challenge, and the authors already do a great job with what's available. But I'd suggest briefly flagging this point in the main text or discussion. A sentence or two noting the inferential nature of the gas identification and perhaps pointing toward future work that could directly probe bubble content, would add helpful nuance without weakening the main conclusions.

Response: We agree with you that the content of the volume inside the bubble features cannot be easily confirmed with the current instrumentation capabilities. In this study, two types of bubbles were discussed: (1) bubbles that exist in the samples prior to imaging are most likely due to gas precipitation; (2) bubbles generated by the electron beam could contain complex contents from radiolysis intermediates, gas products, and water molecules knocked off from the lattice sites. We added two discussions to the manuscript to highlight this aspect.

On page 8, lines 194–196:

In this region, we discovered many trapped gas bubbles in the form of nanoscale cavities in the crystal (Fig. 4a, circular and elliptical features labeled by arrows). Thickness gradients due to bubble curvature are evidenced by the significant contrast variations near the bubble surfaces. The formation of nanosized trapped bubbles is attributed to the relatively fast cooling rate of the thin sample compared to bulk freezing⁴⁷. **Note that these bubbles exist prior to imaging and remain unchanged throughout the imaging process. Therefore, it is most likely that they are generated by gas precipitation instead of radiolysis.**

On page 11, lines 261–263:

Notably, the system reaches a steady state within seconds of gas generation (Fig. 5f), consistent with in situ TEM results. It is crucial to recognize that radiolysis offers one credible explanation for the formation of new bubbles in these experiments, but detecting the content of the gas species is challenging in these experiments. Nonetheless, the formation of substantial H₂ and O₂ species is in agreement with cryogenic EELS analyses of amorphous ice^{58,59}, highlighting the potential role of these gas molecules in constituting bubble volumes in our experiments.

2. In the Main section (line 105), authors refer to “lattice amplitude maps” as a central tool for visualizing and subdomain structure and crystal misorientation. Could the authors consider briefly explaining what a lattice amplitude map represents, how it is constructed and what type of structural information it reveals? A short clarification would help readers better understand who may not be familiar with this analysis technique.

Response: We added a sentence in the main text to briefly clarify the intent of this method (page 4–5, lines 101–102):

To analyze the spatial distribution of the nanoscale defects, we developed a lattice amplitude mapping approach to evaluate the local crystal misorientation from the zone axis (see Methods). Colors in the map represent the relative strength of the lattice patterns in the three $\{\bar{1}100\}$ directions.

We refer the readers to the Methods section for further details. There, a detailed procedure to obtain the lattice amplitude maps can be found (page 22, lines 502 to 510), which we reproduce here:

The lattice amplitude maps were calculated with the following procedures: (1) obtain the Fourier transform image; (2) mask only the circular areas containing the desired pair of lattice reflections and assign zero to all other pixels (mask diameter = 0.2 nm⁻¹); (3) perform inverse Fourier transformation; (4) obtain the absolute value of the image (i.e., flip the sign of negative intensity values); (5) perform Gaussian smoothing with $\sigma = 5$ pixels (this may vary depending on the pixel size); (6) repeat this process for the other two pair of reflections; (7) overlay the three resulting images as a red/green/blue (RGB) stack.

Additionally, would it be possible to distinguish more explicitly how this approach differs from geometric phase analysis (GPA)?

Response: Although the initial steps are similar (masking in Fourier space), the GPA algorithm extracts the phase difference, whereas the “lattice amplitude map (LAM)” method extracts the amplitude. Therefore, GPA provides in-plane lattice distortions that can be used to calculate the strain tensor, while LAM provides an estimate of the out-of-plane tilt of the local domain.

3. In the section discussing MD simulations of low-angle grain boundaries (e.g., Fig. 3), the authors refer to the presence of “clathrate”. Could the authors clarify what is meant by this term in the context of the simulations?

Response: The local configuration of some beads in the MD results, mostly at the interface, are similar to those observed in clathrate hydrates, as identified by the CHILL+ algorithm (ref. 76; *J. Phys. Chem. B*

119, 9369). Therefore, we labeled these beads as “interfacial” in the figures and briefly mentioned that they show “interfacial configurations characteristic of clathrates.” The use of this algorithm has been provided in the Methods section (page 25, lines 567–568), as reproduced here:

The OVITO package⁷⁵ (<https://www.ovito.org/>) was used to perform local symmetry identification using the CHILL+ algorithm⁷⁶ and dislocation analysis using an extended dislocation extraction algorithm (DXA)⁷⁷.